# Facile hermetic TEM grid preparation for molecular imaging of hydrated biological samples at room temperature

Lingli Kong [1,7], Jianfang Liu [1,7], Meng Zhang [1,7], Zhuoyang Lu[2,7], Han Xue[1], Amy Ren[3], Jiankang Liu [2,4], Jinping Li[5], Wai Li Ling [6] ✉ & Gang Ren [1] ✉

Although structures of vitrified supramolecular complexes have been determined at near-atomic resolution, elucidating in situ molecular structure in living cells remains a challenge. Here, we report a straightforward liquid cell technique, originally developed for real-time visualization of dynamics at a liquid-gas interface using transmission electron microscopy, to image wet biological samples. Due to the scattering effects from the liquid phase, the micrographs display an amplitude contrast comparable to that observed in negatively stained samples. We succeed in resolving subunits within the protein complex GroEL imaged in a buffer solution at room temperature. Additionally, we capture various stages of virus cell entry, a process for which only sparse structural data exists due to their transient nature. To scrutinize the morphological details further, we used individual particle electron tomography for 3D reconstruction of each virus. These findings showcase this approach potential as an efficient, cost-effective complement to other microscopy technique in addressing biological questions at the molecular level.

Since the development of the transmission electron microscope (TEM) in the 1930s, imaging biological samples in their aqueous environment has been a main challenge[1,2]. This difficulty arises because the strong interaction between electrons and matters necessitates maintaining the electron microscope column under vacuum to ensure a well-defined trajectory for the imaging electron beam. As a result, any liquid-phase or volatile sample introduced into the microscope will undergo vacuum evaporation, resulting in adverse effects on both the sample and the column vacuum.

Traditionally, TEM studies of aqueous biological samples involve a process of drying and chemical treatment, commonly known as negative staining[3,4]. Notably, the introduction of heavy metal stain

alters the native environment of the protein sample and may even result in the dissociation of protein complexes. While negative staining can improve image contrast and offer an initial assessment of sample quality and shape, it fails to provide an accurate representation of the true structure of biological samples.

In order to study biological samples in their native hydrated state, cryo-electron microscopy (cryo-EM) has been developed[5–8]. This technique involves freezing the samples while preventing the formation of crystalline ice, and then imaging the vitrified samples on sample stages cooled by liquid nitrogen. With direct electron detectors and advanced analysis algorithms, cryo-EM has achieved remarkable success in resolving many three-dimensional (3D) structures of protein

[1]The Molecular Foundry, Lawrence Berkeley National Laboratory, Berkeley, CA 94720, USA. [2]Center for Mitochondrial Biology and Medicine, The Key Laboratory of Biomedical Information Engineering of Ministry of Education, School of Life Science and Technology, Xi'an Jiaotong University, Xi'an, Shaanxi 710049, China. [3]Department of Physics, University of California, Santa Barbara, CA 93106, USA. [4]School of Health and Life Sciences, University of Health and Rehabilitation Sciences, Qingdao, Shandong 266071, China. [5]Department of Biochemistry & Molecular Biology, Mayo Clinic, Jacksonville, FL 32224, USA. [6]Université Grenoble Alpes, CEA, CNRS, IBS, F-38000 Grenoble, France. [7]These authors contributed equally: Lingli Kong, Jianfang Liu, Meng Zhang, Zhuoyang Lu. ✉e-mail: wai-li.ling@ibs.fr; gren@lbl.gov

molecules and complexes at or near atomic resolution[9–11]. Nevertheless, it is nontrivial to work at 200 K below room temperature. Samples can easily act as cold traps and attract contaminations. Achieving stability with the significant temperature difference inevitably requires highly sophisticated and meticulously maintained cryo-EMs. Sample vitrification also requires elaborate machines and accessories. Moreover, it is important to note that the process of vitrification, particularly with the generation of the air-water interface, can potentially distort protein structure or lead to the dissociation of multiunit protein complexes[12]. The ability to resolve and track protein conformations in the liquid state, close to the physiological temperature, would be a significant advancement in structural biology highly complementary to cryo-EM.

In order to image volatile or solution samples at room temperature with TEM, it is necessary to isolate the samples from the vacuum or hermetically encapsulate them. For high-resolution TEM imaging, the sample thickness needs to be less than the meanfree path of the electrons, and the material used to contain the liquid sample must be electron-transparent[1]. In early studies, collodion films[1] were utilized as a barrier between liquid samples and the vacuum. However, the low electron transparency of these films resulted in low-resolution images with only limited structural details[13].

In the past decade, advanced TEM liquid cells using nano-fabricated microchips have emerged and demonstrated remarkable capabilities[14,15]. These microfluidic cells, typically made of silicon nitride, have enabled material scientists to achieve high-resolution real-time TEM imaging of nano-crystal interactions and chemical reactions in liquid environments[16–19]. Additionally, scanning TEM (STEM) has been applied to image entire biological cells at the nanometer resolution[20,21]. However, these sophisticated microchips require specialized holders for mounting[15]. Moreover, the limited accessibility of the nano-fabrication device to produce the microchips has hindered the widespread application in biological sample studies, particularly in preliminary investigations that necessitate extensive fine-tuning of various parameters in the sample preparation process.

An alternative approach to seal liquid samples involves the use of graphene layers deposited on TEM grids[22,23]. In this method, the liquid sample is enclosed between two graphene layers, creating a sandwich-like structure. This technique offers superior resolution since the graphene monolayer is virtually transparent to the electron beam. Moreover, graphene exhibits excellent mechanical strength, high thermal conductivity, and high electrical conductivity, which contribute to greater imaging stability[24,25]. However, graphene is inherently hydrophobic, and there is strong van der Waals attraction between the monolayers. Although surface treatments can mitigate the hydrophobicity, it still remains challenging to control the amount of aqueous solution trapped in the random pockets that formed between the monolayers due to this property.

To facilitate reliable molecular studies of liquid samples using conventional room temperature TEM, we have recently developed a straightforward technique to hermetically encapsulate liquid samples. This method has been successfully applied to capture the real-time dynamics of the liquid-vapor interface structure of liquefied sodium (generated from the electron-irradiated sodium chloride) at nanometer spatial resolution[26]. By employing energy-filtered TEM (EFTEM), we effectively eliminated electrons scattered in the liquid phase, resulting in excellent contrast between solid proteins and their surrounding liquid environment.

In this work, we build upon this approach by adapting the technique for imaging biological samples. We investigate the radiation

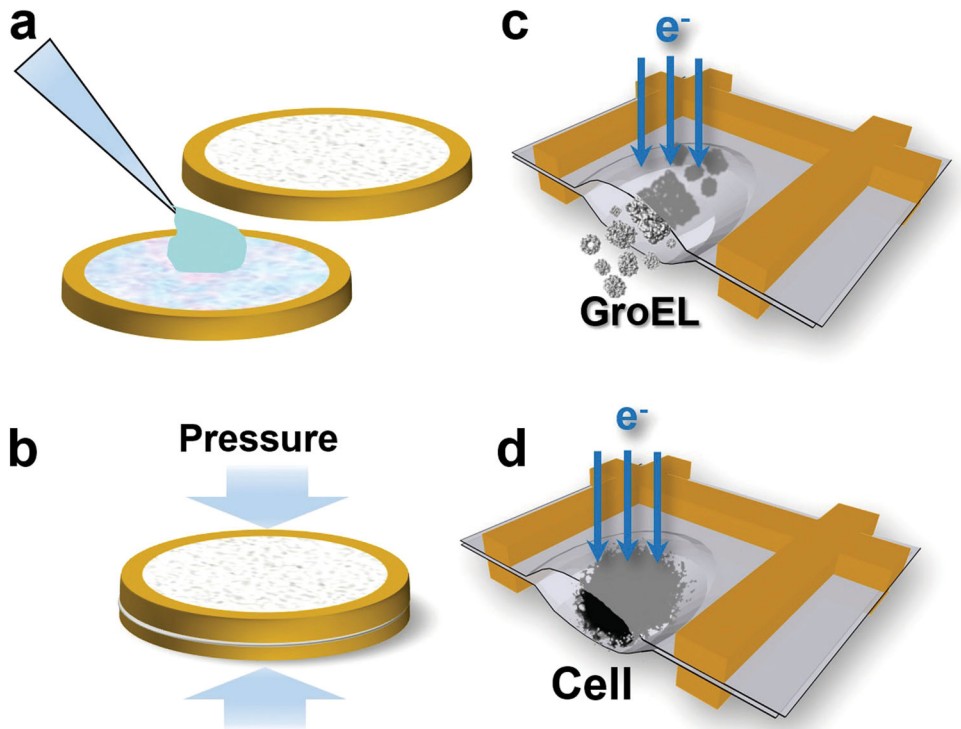

**Fig. 1 | Schematic diagram illustrates the liquid cell assembly and the trapping of liquid samples between the Formvar film in the grid squares for TEM imaging. a** A small aliquot (~0.2 μL) of sample suspension is applied to the center of a Formvar-film coated TEM grid. A second Formvar-film coated TEM grid is carefully aligned with the first grid and placed on top of it under an optical microscope. **b** The grid sandwich is placed in a vise and subjected to controlled pressure applied by a torque wrench. This pressure expels the excess liquid, which is absorbed by filter paper, to obtain the desired thickness of the liquid layer. A small amount of vacuum grease is then applied around the circumference of the grid sandwich to seal the liquid cell. **c** The resulting liquid cell enables TEM imaging of the protein GroEL in its buffer solution. **d** The liquid cell also allows for the imaging of living cells in a wet environment.

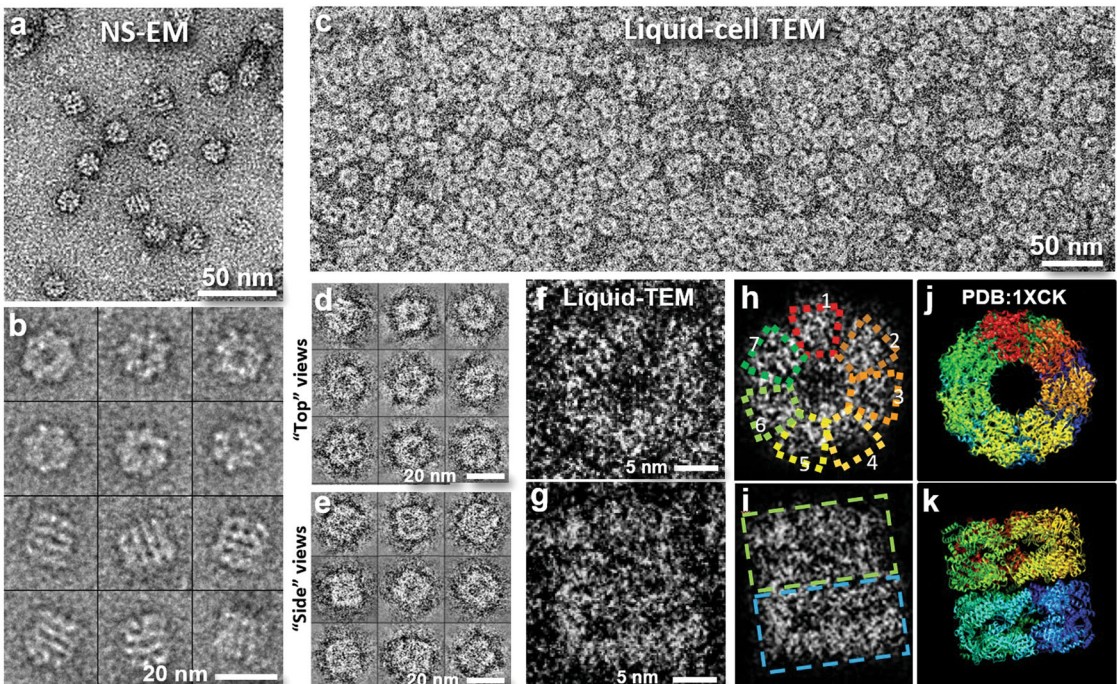

**Fig. 2 | Liquid cell TEM images of GroEL in TBS buffer solution compared to negatively stained electron microscopy (NS-EM) images. a** GroEL sample stained with uranyl formate. **b** Representative particles in NS images. **c** The same GroEL sample (identical batch as used in NS) encapsulated in a liquid cell and imaged at room temperature. The protein particles are uniformly distributed, and the contrast is comparable to that in NS image. Unlike NS, where only molecules attached to the film are retained, all GroEL molecules in the sample volume are present for imaging in the liquid cell. In this experiment, ~5 liquid cell assemblies were prepared and examined, each containing more than 10 grid squares that exhibited a similar density of molecules, showing a consistent contrast, signal-to-noise ratio and appearance. **d, e** Representative GroEL particles in two different orientations ("top" and "side" views). **f, g** Magnified images of two representative particles exhibiting perpendicular views. **h** Image of (**f**) with soft mask, revealing the presence of seven discernible subunits (boxed). **i** Image of (**g**) with soft mask, showing the presence of two stacked rings (boxed). **j, k**, Crystal structure of GroEL (PDB 1XCK) viewed along its $C_7$ symmetric axis (**j**) and $C_2$ symmetric axis (**k**), respectively.

tolerance of liquid samples by monitoring the sample evolution with increasing electron dose. Our imaging results on the model protein complex GroEL demonstrate the ability of liquid cell TEM to resolve nanoscale structural features in solution. We also use a HeLa cell infected with lentivirus as a model system for virus entry. Employing the technique of individual particle electron tomography (IPET)[27] for 3D reconstruction, we identify virus particles fused with the HeLa cell membrane, as well as particles located within the cytoplasm of the HeLa cell. By capturing snapshots of viral cell entry in this system, we establish that our technique can be applied to the studies of transient biological processes, which may lead to insights and hypotheses concerning important biological activities.

## Results

### Liquid cell assembly

We utilized the liquid cell technique that we developed, which has enabled us to observe capillary waves in motion within liquified sodium at nanometer resolution[26], for the encapsulation of biological samples. For this purpose, we employed two standard 300-mesh TEM grids that were coated with Formvar film. These Formvar-coated TEM grids are commercially available, or they can be prepared using straightforward procedures[26]. The grid assembly process is summarized in Fig. 1. Approximately 200 nanoliters of the sample solution were applied to the Formvar film on the first TEM grid under an optical microscope. Subsequently, with the aid of the optical microscope, the second Formvar-coated grid was carefully aligned with the first grid and placed on top of it with the Formvar side facing down. We then applied a well-controlled pressure to the grid sandwich using a precision torque wrench. After several trial runs, we determined that applying 0.55 bar pressure in our setup achieved a thin liquid layer suitable for TEM imaging without crushing the sample. Throughout

these steps, we maintained a controlled relative humidity of >90 % to prevent evaporation of the applied solution volume. Any excess solution from the pressed grid sandwich was removed using filter paper. Finally, we sealed the circumference of the sandwich with vacuum grease and mounted it onto a standard room-temperature TEM sample holder to be introduced into the microscope. Imaging was conducted using a Zeiss Libra 120 kV TEM equipped with an in-column energy filter.

### GroEL imaged in liquid cell

The native GroEL sample in its buffer solution was loaded into a grid sandwich using the aforementioned technique and imaged at room temperature. The thickness of the liquid sample was relatively uniform across the grid (Supplementary Fig. 1). For comparison, the same sample was also prepared by negative staining. The structure of the protein complex GroEL has been previously determined using X-ray crystallography and single-particle cryo-electron microscopy[8]. GroEL is a chaperone protein consisting of 14 monomers with D7 symmetry, a molecular mass of ~800 kDa, and dimensions of ~14 × 14 × 14 nm³. Each monomer has dimensions of ~6 × 6 × 7 nm³, as reported in ref. 8.

Figure 2 provides a comparison between the negatively stained GroEL sample (Fig. 2a–b) and the same GroEL sample imaged in buffer solution, encapsulated using our developed technique (Fig. 2c–g). In Fig. 2c, the GroEL particles are evenly distributed, which accurately represents their solution state. Notably, the liquid cell sample demonstrates image contrast comparable to that of the negatively stained sample. The signal-to-noise ratio (SNR) for the particles in the liquid cell is 1.05, compared to 2.56 for negatively stained proteins (Supplementary Table 1). The similar image contrast (reverse of that in cryo-EM) indicates that the liquid phase has

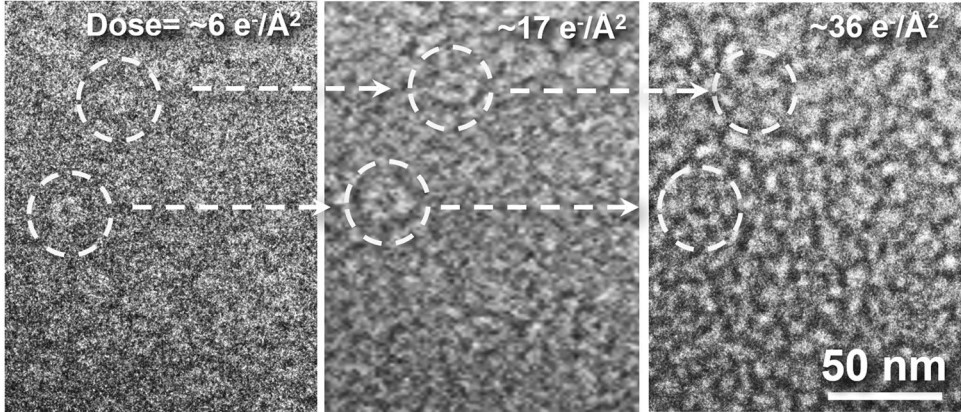

**Fig. 3 | Evolution of the GroEL sample in liquid cell as a function of electron dose.** Impact of electron irradiation is studied by controlling the electron dose on the sample using the low-dose imaging technique. Two GroEL molecules in the same sample region are circled. Radiation damage is clearly apparent at 6 e⁻·Å⁻², 10x less than the conventionally acceptable dose for high-resolution cryo-TEM imaging. The two molecules are barely distinguishable from the background after an exposure of 17 e⁻·Å⁻², and then vanish into the background at 36 e⁻·Å⁻². The radiation damage experiments have been conducted on ~5 liquid cell assemblies with 10–20 window areas. The radiation damage process has shown similar evolution across all window areas.

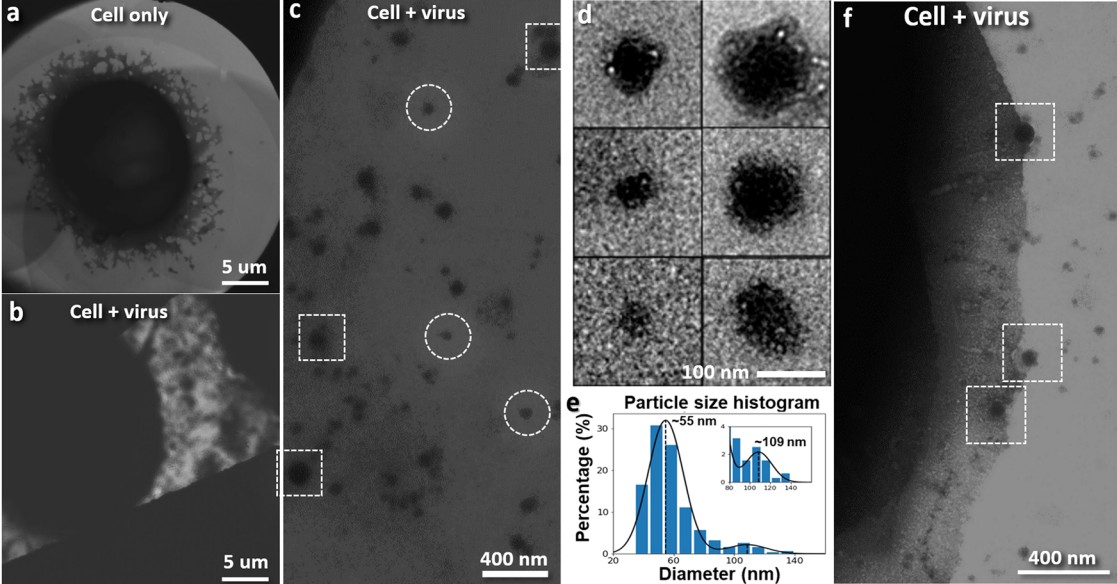

**Fig. 4 | Liquid phase TEM images of HeLa cells in growth medium in the absence (cell only) and presence of lentiviral vectors (cell + virus). a** Representative image showing a HeLa cell in a growth medium without lentiviral vectors. ~20 HeLa cells have been examined on 4 grid assemblies. Clear background with no nanoparticles has been observed around the cells. **b** Representative image showing a HeLa cell in growth medium incubated with lentiviral vectors. ~20 HeLa cells have been examined on 4 grid assemblies. Abundant nanoparticles are found in the background. **c** Magnified image of the surroundings of a HeLa cell mixed with lentiviral vectors. Part of the HeLa cell can be observed in the upper left corner. Numerous small nanoparticles and larger particles (framed in circles and squares, respectively) are visible. **d** 315 nanoparticles found in the background around HeLa cells mixed with lentivirus are imaged. Representative images of small (~50 nm) nanoparticles (left) and large (~100 nm) particles (right). Some smaller nanoparticles reveal similar surface features found on larger particles. **e** Histogram illustrating the diameter of 315 particles, exhibiting a major peak at ~55 nm and a minor peak at ~109 nm. The size distribution of the larger particles is coherent with that of lentiviral vectors. The more abundant smaller particles may comprise vesicles released by the cells during infection. **f** Image (after high-pass filtering) of the edge of a HeLa cell that has a relatively smooth plasma membrane in the sample incubated with lentivirus. Four nanoparticles consistent with the size of lentivirus are found attached to the edge of the cell. Three of these potential sites of virus entry are boxed.

a stronger ability to scatter electrons compared with solid proteins. In our experimental setup, this effect was particularly pronounced due to the elimination of scattered electrons using zero-loss EFTEM.

However, unlike the opaque surface features observed in negatively stained images (Fig. 2a–b), which are defined by the heavy metal stain coating, the liquid cell images provide projections of the sample that offer insights into the internal structure within the 3D molecules. These images allow for the identification of top views and side views of the protein in projections (Fig. 2d, e) similar to those seen in cryo-EM images[8]. Magnified images of representative particles presented in their top view reveal the seven discernible subunits (Fig. 2f, h), similar to the structure determined by X-ray crystallography (Fig. 2j)[28]. Side views (Fig. 2g, i) also exhibit a domain structure consistent with that determined by X-ray crystallography (Fig. 2k)[28]. Although the nanometer resolution achieved in our experimental setup is not adequate for satisfactory 3D reconstruction, the level of structural details directly observed in the liquid sample surpasses what has been achieved using other existing techniques for wet samples.

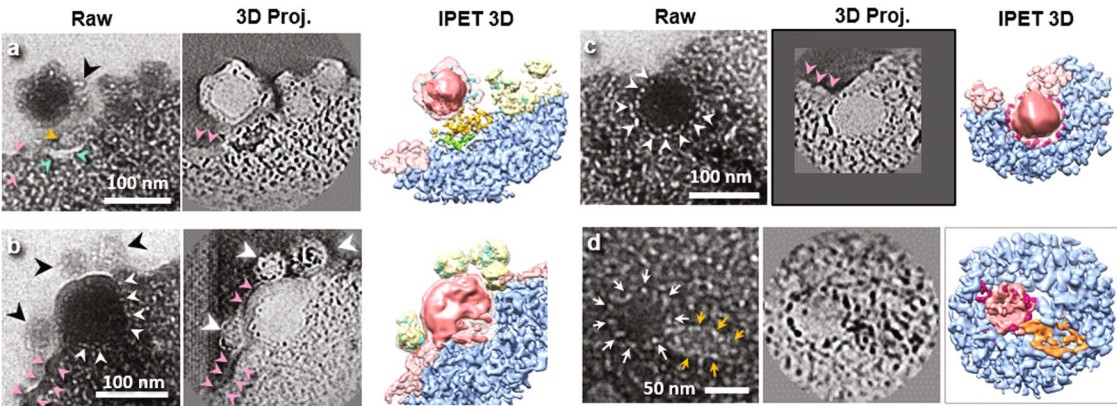

**Fig. 5 | Liquid phase TEM images and 3D reconstruction of lentiviral vectors interacting with HeLa cell membrane.** Tilted image (Raw), contour enhanced projections (3D Proj.), and IPET 3D reconstruction of the three boxed areas in Fig. 4(f) are shown. **a** Particle attachment (~7.2 nm resolution): The cell plasma membrane exhibits a concave surface in response to viral attachment. Orange and cyan arrows indicate two types of structures connecting the virus-like particle and the plasma membrane. One of the protrusions found on the viral surface is highlighted by the black arrow. The pink arrows mark a dense and thick membrane region (see Supplementary Fig. 4), potentially representing a lipid raft. **b** Half-embedded particle in the cell membrane (~7.4 nm resolution): The outer shell of this virus-like particle lacks the protrusions observed in (**a**). Conversely, three small nanoparticles (black arrows) with diameters of ~50 nm are connected to the virus-like particle and appear to be coated with features resembling the protrusions observed in (**a**). A tight delimited coat of lighter density separates the half virus from the HeLa cell (white arrows). Another region of abnormally dense membrane is indicated by pink arrows. **c** Embedded particle (~6.5 nm resolution): No significant difference is observed between the embedded particle and the half-embedded particle; the particle remains round with a tight-fitting coat of lesser density. The pink arrows once again indicate a region of dense cell membrane at the cell surface. However, no protrusions are found on the embedded virus surface. **d** Endocytosed particle (~6.4 nm resolution): The round particle (white arrows) is seen attached to a cone-like structure (orange arrows).

## Estimation of radiation tolerance

The GroEL images presented in Fig. 2 were acquired using a low-dose imaging approach, with a total dose of ~1.0 e⁻Å⁻². In low-dose imaging, the image acquisition areas between high-magnification and low-magnification settings are aligned using stage and image shifts. The searches for areas of interest are conducted at low magnification, where the electron dose applied to the sample is negligible. Focusing is performed in a distinct area separated from the data collection area, ensuring that the latter remains unexposed. The region of interest is exposed to a significant and well-calibrated dose (in this case ~1.0 e⁻Å⁻²) only during the high-magnification image acquisition. By employing low-dose imaging, we were able to closely monitor the sample under the influence of electron beam irradiation.

Significantly, a degradation in image quality was already noticeable when the cumulated dose reached ~2.9 e⁻Å⁻². At a cumulated dose of 6 e⁻Å⁻² (Fig. 3), the GroEL molecules were barely distinguishable from the background. It is worth noting that this dose is roughly ten times lower than the general acceptable limit for vitrified protein samples in cryo-EM[29]. Such a limitation is anticipated in room temperature imaging, as the low temperature in cryo-EM is known to slow down the detrimental processes induced by the electron beam[29].

## Liquid cell TEM applied to virus entry

Subsequently, we investigated the potential of this approach to shed light on molecular mechanisms in living systems, particularly virus entry. Virus entry is a crucial biological process with significant medical implications. It represents an attractive target for antiviral drugs due to the extracellular nature of the entry machinery, which facilitates the access of drug molecules. Furthermore, comprehending virus entry is crucial to efficiently engineer viruses that can selectively infect specific cells (transduction retargeting), particularly when using viruses as delivery vehicles for drugs or genetic material. Despite the importance of understanding virus entry, limited in situ structural information is available currently, mainly due to the transient nature of the process[30,31].

We utilized our technique to encapsulate a model system of virus entry, which consisted of HeLa cells mixed with a lentiviral protein transfer vector capable of infecting HeLa cells. Lentiviral vectors are virus-like particles derived from the human immunodeficiency virus type 1 (HIV-1) and are employed as recombinant vaccine vectors to deliver gene sequences encoding antigen of pathogens into cells[32]. In our experiments, the lentiviral vector used was pseudo-typed with the G-glycoprotein of vesicular stomatitis Indiana virus (VSV-G), enabling infection of a wide range of cells, including HeLa cells[33]. This vector supports only a single round of infection and does not undergo replication.

Figure 4 displays images of the HeLa cells (Fig. 4a) and the cells mixed with lentiviral vectors in the cell culture medium (Fig. 4b–c). The background solution in the latter shows numerous particles with varied diameters but similar surface features. These particles are not present in the control sample containing only HeLa cells (see Supplementary Fig. 2). Statistical analysis reveals two distinct populations of particles (Fig. 4d–e). The smaller particles exhibit a peak diameter of ~55 nm, while the larger particles have a peak diameter of ~109 nm. The size distribution of the larger particles aligns with the size range of lentiviral vectors (64 to 170 nm) observed by negative staining[34] and cryo-EM[35]. The larger population of smaller particles could comprise extracellular vesicles released by the stressed cells, playing a role in intercellular communication[36]. Figure 4f shows a cell with several larger particles adhering to its cell surface, possibly representing different stages of cell entry (Supplementary Movie 1). Smaller particles are present in the vicinity of these adhesion sites. These particles could possibly originate from the additional membrane generated during virus fusion (see discussion section below and Supplementary Fig. 3) and may be part of the population of background particles observed. We also observe that the plasma membrane near the viral entry areas appears denser than the membrane away from these sites (Supplementary Fig. 4). Similar observations have been reported in negatively stained images of virally infected cells[37–40].

## IPET of virus cell entry

To confirm the 3D positioning of these particles (Fig. 4f) in relation to the cell membrane, we conducted electron tomography on the specific region of the sample. Given that the size and characteristics of these particles were consistent with those documented for lentiviral vectors in negative staining and cryo-EM studies, we hypothesized that these

particles were indeed lentiviral vectors. Figure 5 presents magnified views of the particles obtained from the zero-tilt projection of this sample region, as well as the projections derived from the 3D reconstruction and its density maps (the superposition of the positive and negative density maps, which highlight features that scatter electrons more and less than the background, respectively), following data processing with IPET (Supplementary Fig. 5–7). The resolution of the maps, estimated by the Fourier Shell Correlation (FSC) of 0.5, showed that the maps had a resolution better than 10 nm (Supplementary Fig. 5–7). The 3D reconstruction confirms that the virus-like particles are indeed in direct contact with the cell membrane, which appears to undergo deformation to accommodate the curvature of the particles. These observations suggest that the particles are in the process of entering the cells, considering that the lentiviral vector used in the experiment was designed not to replicate.

By examining the relative contact area between the virus-like particles and the cell membrane, we observed a range of particles that may correspond to different stages of cell entry. In Fig. 5a, we see a particle with a diameter of ~104 nm, seemingly connected to the cell surface through protrusions on its surface. The associated cell membrane forms a concave surface that partially envelopes the particle covering around a quarter of its surface. Similar concave surfaces have been reported in the case of VSV-G (GFP labeled pseudo-typed vectors) infecting T-cells, as observed by negative staining TEM[37]. Additionally, apart from the protrusions (indicated by a black arrow in Fig. 5a), other structures can be observed bridging the large particle and the cell surface (indicated by cyan and orange arrows in Fig. 5a).

Figure 5b shows another spherical particle, ~130-nm in diameter, which is consistent with the size of a lentiviral vector, partially embedded within the cell membrane. The region of the particle outside of the cell membrane displays a smooth surface. Unlike the particle in Fig. 5a, the small protrusions observed previously (black arrow in Fig. 5a) are absent on the surface of this particle. However, intriguingly, features resembling the absent protrusions can be observed on three vesicle-like nanoparticles that are attached to the viral particle (indicated by the black arrows in Fig. 5b). These three nanoparticles have diameters of ~47 nm, ~64 nm, and ~65 nm, respectively.

In Fig. 5c, another virus-like particle is shown immediately after being embedded into the cytoplasm. The plasma membrane is observed to be sealed above the embedded particle. This particular particle has a slightly smaller diameter (~96 nm) compared with the particles observed in Fig. 5b. Notably, the spherical particle exhibits a continuous coat of tiny particles (indicated by white arrows in Fig. 5c), which appear to be in direct contact with the cytoplasm.

Additionally, we observed particles in the cytoplasm that appeared to be internalized viral particles (Fig. 5d, Supplementary Fig. 8–10 and Supplementary Movie 1), positioned more than ~100 nm from the plasma membrane surface. Figure 5d illustrates one of these particles (Supplementary Fig. 9). The particle contains two distinct parts: a globular portion (with a diameter of ~70 nm, indicated by white arrows in Fig. 5d) attached to a cone-shaped portion (measuring ~30 × ~45 nm, indicated by the orange arrows).

## Discussion

The findings of this study highlight the effectiveness and simplicity of the liquid cell technique developed for the examination of wet biological specimens at the molecular level under room temperature. The technique allows for achieving resolution beyond 10 nm, as demonstrated in the GroEL images and through FSC estimation in the tomograms. Notably, this approach enables the examination of whole-cell samples in the culture medium at room temperature, without the need for staining, air-drying, or plastic embedding, thereby avoiding associated artifacts. The assembled liquid cells can be easily mounted on a standard TEM holder, facilitating large-scale screening. Importantly, this technique provides true projections of the biological

samples within their native environments, offering volume information with significantly enhanced contrast than cryo-EM.

When conducting liquid cell TEM, it is essential to exercise caution and adhere to certain precautions. Inadequate sealing of the liquid cell can lead to sample leakage, resulting in a degradation of the column vacuum. If a high volume of liquid cell experiments is to be performed, it is preferable to have an additional pumping system at the sample level (similar to those commonly found in TEMs designed to accommodate cryo-EM experiments) to minimize potential contamination of the microscope. The choice of glue types and sealing procedures significantly influences the quality of the seal. Additionally, for optimal image quality, various factors should be considered, including the materials used for the film, the size of the windows (grid mesh size), the thickness of the liquid layer, and the dose of illumination. Our findings emphasize the extreme sensitivity of liquid samples to radiation damage. Therefore, employing a high-speed, high-sensitivity detector is recommended for high-resolution imaging using liquid cell technique.

We have demonstrated that liquid cell TEM imaging of biological samples exhibits contrast comparable to negatively stained samples (see Supplementary Table 1), which is significantly higher than in cryo-EM samples. This enhanced contrast is particularly notable when coupled with EFTEM. The increased scattering of electrons in liquid samples can be attributed to the greater freedom of water molecules at room temperature compared to the rigid structure of amorphous ice. The presence of additionally available vibrational modes in liquid (*c.f.* phonons in solids) enables a wider range of energy exchange with incoming electrons. Additionally, in a liquid environment, electrons are more likely to encounter constantly moving molecules, as opposed to stationary molecules in a solid. Consequently, liquid water scatters electrons to a greater extent than solid samples composed of elements with similar atomic weight[41,42]. However, incidental solution concentration during liquid cell preparation may also have contributed to the extra scattering; despite taking precautions, evaporation can possibly occur, especially considering the small volume applied. In such a case, the concentration of salt in the growth medium may increase, leading to additional scattering. Regardless of the cause, the effect becomes more prominent when using EFTEM, as it eliminates scattered electrons from the image.

The preliminary findings from our investigation of HeLa cells with lentiviral vectors have revealed potential stages of viral cell entry as illustrated in Fig.5a–d. While emphasizing the fact that the present dataset is a small and preliminary dataset, which is statistically insignificant for any appropriate claim, we suggest a highly speculative pathway (depicted in Supplementary Fig. 12) for VSV-G initiated virus entry based on these preliminary observations as a proof-of-principle of what the technique may be capable of delivering in cellular EM.

While we are currently unable to confidently assign protein densities to those seen in Fig. 5a–d, we hypothesize that the protrusions (black arrows indicated in Fig. 5a) could conceivably represent the viral surface spike proteins. It is known that VSV-G glycoprotein interacts with cell receptors during the process of cell entry, as exemplified by its binding to low-density lipoprotein receptors (LDL-R)[43]. The additional structures observed (indicated by cyan arrows and orange arrows in Fig. 5a) could potentially correspond to cell receptors and co-receptors[44] that initiate the membrane fusion process. However, the confirmation of these speculations would obviously necessitate further experiments, and these interpretations should be treated with caution.

Additionally, we hypothesize that the nanoparticles observed in Fig. 5b could be liposomes formed by the virus membrane encompassing spike proteins during the destabilization of the cell plasma membrane as the virus-like particle penetrates the cell, however, we must caution the reader that we cannot rule out that nanoparticles may be derived from infected cells shedding membrane, as opposed to the

nanoparticles being a by-product of infection. Despite the fact that ~300 nanoparticles were observed in the solution around the cell, none of the 15 virus-like particles in the cellular solution were found attached to nanoparticles that have similar surface features. In contrast, two out of the three virus-like particles affixed to the cell surface were found attached to this type of nanoparticle. It is known that lipid rafts, which play a role in virus entry by localizing cell receptors for viral entry, can reduce the fluidity of the cell membrane[45]. If the viral membrane were to merge with the plasma membrane, we would expect to observe features such as wrinkles or alterations in the membrane surface area. However, such features were not observed in our (albeit small) dataset. In fact, a simple estimation indicated that the fused viral/cell membrane had a similar surface area to the total surface area of the three nanoparticles (Supplementary Fig. 3). Therefore, we propose that liposomes may be formed from the excess lipids of the viral membrane during virus entry. Whereas extracellular vesicles released by the stressed cells may have contributed to the main population of the small particles in the solution, which was not observed in the solution of the HeLa cell sample without the lentivirus (Supplementary Fig. 2), this hypothesis suggests that nanoparticles from excess viral membrane may also have contributed to the abundance of small particles. More experiments will be required to determine whether the observed nanoparticles are indeed formed from viral infection, or are instead released by infected cells post-infection. Further experiments to collect a statistically significant data set will be essential to verify the hypothesis.

If the nanoparticles observed at the site of virus entry are indeed liposomes formed by the excess lipids resulting from the fusion of the plasma membrane and the viral membrane (Supplementary Fig. 3), the matrix protein may come in direct contact with the cytoplasm upon entry. Further investigations using higher-resolution imaging techniques, such as cryo-EM with direct electron detectors, or immunolabelling, will be necessary to confirm these findings. Additionally, the potential presence of liposomes containing spike proteins and its implications in the host immune response warrant further investigations.

This study showcases the potential of a straightforward hermetic TEM grid preparation method as a valuable and cost-efficient technique for molecular imaging of hydrated biological samples at ambient temperature. This approach complements other microscopy techniques and presents a promising solution for exploring crucial biological questions at the molecular level. Although our current observations are insufficient to draw any definitive biological conclusions, the demonstration of the potential of this methodology serves as a proof-of-principle on its prospective applications.

## Methods

### Biological sample preparation

The lentivirus was a third-generation virus packaged with three co-transfection vectors, plasmids *pLKO.1*, *psPAX2*, and *pMD2.G*, in which *pMD2.G* contained the genes encoding envelope proteins, specifically, the G glycoprotein of vesicular stomatitis Indiana virus (VSV-G). The presence of the VSV-G protein enabled the lentivirus to infect a wide range of cells, including HeLa cell[46]. HeLa cells (CCL−2T™) were ordered from American Type Culture Collection (ATCC, Manassas, Virginia, USA), which were cultured in Gibco minimum essential media (MEM, containing 10% fetal bovine serum, FBS) at 37 °C with 5% $CO_2$. The prepared virus-infected HeLa cells were conducted by incubated with lentiviral vectors at a ratio of 1:50 (1 cell to 50 virus) for a duration of 12 h. The sample of GroEL used for negative staining and liquid cell TEM imaging was provided by Dr. Scott Stagg's laboratory. The protein was present at a concentration of ~0.5 mg/mL in Tris-buffered saline (TBS, i.e. 50 mM Tris, pH 7.4, 50 mM KCl, and 1 mM DTT). For negative staining, the sample was diluted and stained by following the published optimized negative-staining (OpNS) protocol[4]. Subsequently, it

is imaged using a Zeiss Libra 120 TEM (Carl Zeiss NTS) equipped with a Gatan UltraScan 4k × 4k charge-coupled device (CCD) detector.

### Liquid-phase TEM specimen preparation

The method used for sandwiching the biological samples followed the published protocol[26]. In brief, ~0.2 µl of the native liquid sample solution, without any staining or labeling, was encapsulated between two layers of Formvar deposited onto two 300-mesh TEM copper grids (~3 mm in diameter, ~100 µm in window width) at room temperature with a humidity level of >90%. The two TEM grids were then pressed together under a pressure of ~8 psi (~0.55 bar) for ~20 s. Any excess solution was carefully removed using filter paper. Next, the edge of the compressed grids was sealed using high-vacuum grease (DuPont Molykote, USA Lab). Finally, the grid sandwich was mounted onto a regular TEM holder for further analysis.

### Optimizing TEM imaging

The liquid phase TEM grids, which contained label-free biological samples, were examined using a Zeiss Libra 120 Plus TEM (Carl Zeiss NTS) at room temperature. The TEM was operated at a high tension of 120 kV with a 20 eV slit for zero-loss in-column energy filtering. Micrographs were acquired using a Gatan UltraScan 4k × 4k charge coupled device (CCD) with a defocus up to 12 µm and magnification from 1000× to 80,000× (corresponding to pixel sizes of 107 Å to 1.48 Å in the specimen). To assess radiation damage, the GroEL sample was subjected to an illumination dose ranging from ~1 to ~36 e⁻Å⁻² under magnification of 40,000× to 80,000× and defocus up to −3 µm. Approximately 300 micrographs were acquired under low-dose conditions using a magnification of 80,000x and a defocus range of ~0.1 to ~1.0 µm. For the HeLa cells, approximately 100 micrographs were acquired under low-dose conditions at a magnification of 1000× to 20,000×.

### Electron tomography tilt series acquisition

For electron tomography tilt series acquisition, the sample holder containing the grid of wet HeLa cells was tilted at a series of angles ranging from −36° to +60° in steps of 1°. The tilt angles were controlled by both Gatan tomography software and fully mechanically controlled automated electron tomography software[47], which were preinstalled in the microscope (Zeiss Libra 120 TEM). During the tilt series acquisition, the TEM was operated at 120 kV with a 20 eV slit for zero-loss energy filtering. The images were captured using a Gatan Ultrascan 4k × 4k CCD. The defocus was set to ~11 µm, and the magnification used was either 1000× or 10,000× (corresponding to 107 Å and 10.7 Å per pixel, respectively, in the specimen). The total electron dose for a complete tilt series was ~30 e⁻Å⁻².

Due to the extensive overlapping of cell surface membranes, it was challenging to accurately discern the presence of virus particles near the boundary of the cell surface. However, we were able to locate a rectangular-shaped cell measuring ~17 µm × ~6 µm in dimension (Supplementary Fig. 11 and Supplementary Movie 2) that exhibited a relatively smooth surface membrane. Within this cell, two chain-shaped densities were observed near the center (Supplementary Fig. 11 and Supplementary Movie 2). During the tilt series acquisition at low magnification (~1000×), these densities exhibited consistent shape and size, indicating that they were unlikely to be artifacts from radiation damage but rather genuine features within the cell. The subsequent 3D reconstruction using IPET confirmed the presence of two chain-shaped densities with similar size and shape near the center of this rectangular cell (Supplementary Fig. 11c and Supplementary Movie 1 and 2). The overall morphology observed in this rectangular cell closely resembled that of a dividing cell with two copies of chromosomes, as reported in images from light microscopy[48].

## Image preprocessing

The defocus and astigmatism of each micrograph were measured and the contrast transfer function (CTF) of each image was corrected using EMAN CTFit software[49]. Prior to CTF correction, X-ray speckles were eliminated, and micrographs with noticeable drift were excluded from the analysis. To enhance high-resolution details and to remove background noise, a Gaussian boundary high-pass filter was applied to each micrograph within a resolution range of 200 nm to 500 nm. The micrographs in the tilt series were initially aligned using the IMOD software package[50]. Subsequently, the CTF was corrected using TOMOCTF[51]. The tilt series of virus-like particles were then semi-automatically tracked and cropped into square windows of 128 to 360 pixels (corresponding to ~137 to ~386 nm) using the IPET software[27].

## Statistical analysis of virus-like particle sizes

For the statistical analysis of particle sizes, all particles surrounding the cell (a total of 315 particles) were selected and windowed. The area of each particle was measured by Python Open CV. The histograms of the particles with diameters above and below 90 nm were fitted by Gaussian curves. The curves were merged based on their respective particle numbers using the curve_fit function from the scipy.optimize module in Python.

## IPET 3D reconstruction

The particles of interest (HeLa cells, viruses, and nanoparticles) in the tilt series were reconstructed using the IPET software[27]. In brief, a circular mask with a Gaussian edge was applied to each image, followed by a 3D reconstruction through an iteration refinement process using a series of soft-boundary masks and filters. To visualize the objects with positive or negative contrast relative to the background, a superimposed 3D density map was generated by combining the positive contour map of the final IPET 3D density maps with its negative contour map using the Chimera software[52]. The resolution of the final 3D map was estimated based on the intra-FSC analysis[27]. Specifically, the aligned images were divided into two groups based on their tilting index (odd- or even-numbered) to generate two 3D reconstructions. An FSC curve was computed using these two reconstructions, and the frequency at which the FSC curve fell to a value of 0.5 was used to indicate the resolution.

## Reporting summary

Further information on research design is available in the Nature Portfolio Reporting Summary linked to this article.

# Data availability

The data that support this study are available from the corresponding authors upon request. The liquid TEM density maps of three viruses are available from the EM databank as EMD-25890 (virus 1), EMD-25894 (virus 2) and EMD-25895 (virus 3).

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

## Acknowledgements

We express our gratitude to Dr. Jianying Zhang for providing the viruses and cells, Dr. Scott Stagg for providing the GroEL sample, Dr. Lei Zhang for early testing, Matthew J. Rames, Jacob Jay, Gumaro Contreras, Maya Tome, and Teresa Fowler for discussions and comments. Work at the Molecular Foundry was supported by the Office of Science of the U.S. Department of Energy under Contract No. DE-AC02-05CH11231. G. Ren, L. Kong, Jianfang Liu and M. Zhang were partially supported by the National Heart, Lung, and Blood Institute (NHLBI), the National Institute of Mental Health (NIMH) and the National Institute of Diabetes and Digestive and Kidney Diseases (NIDDK) of the National Institutes of Health under award numbers of R01HL115153, R01MH077303, and R01DK042667. Jiankang Liu was supported by the National Natural Science Foundation of China Integrated Project of the Major Research Plan 92249303 and General Project 32171102. W.L. Ling and G. Ren acknowledge the support of the France-Berkeley Fund (FBF Activation 2022 #26_2022, LBNL). W.L. Ling also acknowledges the support of Dr. G. Schoehn from the IBS electron microscopy platform and thanks Dr. G. Effantin for the insightful discussion. IBS acknowledges integration into the Interdisciplinary Research Institute of Grenoble (IRIG, CEA).

## Author contributions

This project was initiated and designed by A. R. and G.R. HeLa cell samples were prepared by J. Li. The images of cells and viruses were acquired by Jianfang Liu, A. R. and G. R. The GroEL images were collected by Z. L. The images were preprocessed by L. K., Jianfang Liu, M. Z. and G. R. The data were interpreted and analyzed by L. K., M. Z., Jianfang Liu, Jiankang Liu, W. L. L. and G. R. The figures were prepared by L. K., A. R., W. L. L. and G. R. The movies were produced by L. K. The manuscript was written by W. L. L. and G. R., with contributions from A. R., L. K., M. Z., Jianfang Liu, H. X., and J. Li.

## Competing interests

The authors declare no competing interests.
