## [Peer Review File · Nature Communications]

REVIEWER COMMENTS

Reviewer #1 (Remarks to the Author):

The manuscript by Kong et al. reports the use of a very simple liquid cell to observe biological structures in a solution environment, including protein complexes, cells, and viruses, as well as the interaction between cells and viral particles at various stages of attachment, endocytosis, and capsid degradation. The approach and the results are novel in the field of liquid-cell TEM. The examples used — protein complexes, cells and viral particles that can infect those cells — are a good choice and the comparison to negative stain TEM and illustration of the e-beam effects is important for demonstrating the utility of the method. The methodology is sound and the approach and results are clearly articulated. My opinion is that publication of these achievements is important for the field and, thus, the work should be accepted in Nature Comm. once certain revisions have been made.

First, it is difficult for me to believe that the only reason this approach is useful is because not everyone can afford to do high end cryoTEM and, consequently, this “poor man’s” version provides an alternative. Yet, that is essentially the argument made in the introduction. The importance of the work needs to be better articulated.

Second, I find the discussion of the nanoparticles seen in Fig. 4 to be uninformative, particularly those that make up the dominant peak at 55 nm in the size distribution. What are they or, at least, what could they be and why do they far outnumber the viral particles? If these are not seen in cryoTEM imaging or negative stain imaging, then they must be an artifact of sample preparation. The fact that the nanoparticles seen in the images of Fig. 5., particularly the ones attached to the cell in Fig. 5a and those being related to viral protrusions in Fig. 5b., have the same diameter as the nanoparticles that make up the peak in the size distribution makes this an important point to adequately address. The paper would be strengthened if this unfortunate aspect of the samples were resolved or at least explained.

Third, I am fairly certain the data in Fig. 5 are over interpreted. Identification of viral spikes, and membrane thickening, protein-lipid nanoparticles and uncoating of the endocytosed particles looks to me to be beyond what is a fair interpretation. The Discussion goes even further and admittedly presents a speculative model, concluding, “The proposed entry pathway differs from current conceptions of virus entry...” I am a proponent of speculating on what experimental results might apply, as long as the authors make it clear that the statements are, indeed, speculative. However, for the non-expert, the speculation presented here seems to be an extreme leap of faith. But, I am not a biologist; it may well be that an expert in cell biology and viral infectivity would look at those images and see the interpretation as obvious. I will leave it to other reviewers to make this assessment.

Finally, although the manuscript is generally well written and easy to comprehend, there are numerous minor grammatical issues that need to be addressed.

Reviewer #2 (Remarks to the Author):

This is a manuscript by the Ren lab, focused on applying the IPET technique to biological specimens enclosed in a liquid environment. The sample preparation technique is easy to use as described, helping to make liquid-EM studies more accessible to the community. The samples were nicely selected using single particle techniques to evaluate gro-el and lentiviruses to infect hela cells. I recommend its publication pending a few minor adjustments.

- Can the authors better define the term “molten sodium” a bit more clearly? I understand they have a publication on this technique already, but for the uninformed reader who may be looking for a concentration measurement, it would be helpful.

- Can a few very low magnification images be provided in figure 1 or be added to the supplemental materials for the reader to get the sense of liquid coverage and differences in liquid thickness?

- The zero-loss EF-TEM images look great, well done!

What is the signal-to-noise ratio of representative images of gro-el in liquid compared with NS? This would help to quantify the amplitude contrast and numerically support the claims of similar contrast.

-In figure 2f and g, its a bit hard to see the seven domains, can they be numbered on the panels to help the reader distinguish the features?

- Have you thought about doing 3D reconstructions of the gro-el data? Seems like you would achieve a nice structure....please comment whether you think this is possible, or why it may not be feasible (limited views, etc)...

- Is the liquid thickness in these enclosures dictated by the diameter of the sample particles? If so, what's the estimated thickness of each type of sample (particles vs viruses)...I like the virus entry studies. Albeit speculative, it brings a new technique to the field that one can optimize for future lines of investigation.

- Minor technical note, I was unable to open the video files, but I did see the supplemental materials.

Reviewer #3 (Remarks to the Author):

This reviewer would mainly like to comment on the virus entry aspect of this manuscript. It is an impressive feat that authors could observe the change in morphology of the virus during infection using the liquid cell system. However, the authors' proposed detailed description of the entry process is highly speculative and contradictory to the current consensus in the field on the viral entry process. The authors should tone down the biological claims significantly until they have very strong evidence – immune-staining, statistics etc. Without additional experiments, the authors should just describe the general viral entry process, and not point to any proteins and receptors or speculate in detail about mechanisms of viral entry etc.

Major comments:

(1) The authors comment on the spike and M proteins by pointing out negative (white) densities, but if those are supposed to be proteins, so why would they have negative densities? It is probable that the negative densities are bubbles caused by radiation damage to the sample, and thus it is best not to make claims like this.

(2) A lot of these observational claims must be supported by statistics, e.g. how often do the liposomes associated with viruses appear during entry or they are just random?

(3) The authors speculate that the liposomes are parts of the viral membrane being released during the virus entry process by comparison to uninfected cells. However, infected cells are generally sicker than uninfected cells, with a more variable morphology as a result. How would the authors know if the liposomes are derived from the virus and not shedding of membrane from the sick cells?

(4) The authors should provide statistics about the claimed thick cell membranes and their localization in regions around virus and not around virus.

(5) Statistics are required for how often conical particles are associated with globular particles (capsid and matrix, respectively, as claimed by the authors) post-viral entry.

This reviewer thinks that the authors should focus on the liquid cell method and showing that it is applicable to the study of both purified protein samples and cells infected with viruses. A deeper examination of speculated entry mechanisms based on the data in the manuscript is out of the scope of

this manuscript and could be highly controversial and would need to be supported by orthogonal experiments that were not shown here.

REVIEWER COMMENTS and AUTHOR RESPONSES

Reviewer #1 (Remarks to the Author):

The manuscript by Kong et al. reports the use of a very simple liquid cell to observe biological structures in a solution environment, including protein complexes, cells, and viruses, as well as the interaction between cells and viral particles at various stages of attachment, endocytosis, and capsid degradation. The approach and the results are novel in the field of liquid-cell TEM. The examples used — protein complexes, cells and viral particles that can infect those cells — are a good choice and the comparison to negative stain TEM and illustration of the e-beam effects is important for demonstrating the utility of the method. The methodology is sound and the approach and results are clearly articulated. My opinion is that publication of these achievements is important for the field and, thus, the work should be accepted in Nature Comm. once certain revisions have been made.

Comment #1.1: *First, it is difficult for me to believe that the only reason this approach is useful is because not everyone can afford to do high end cryoTEM and, consequently, this “poor man’s” version provides an alternative. Yet, that is essentially the argument made in the introduction. The importance of the work needs to be better articulated.*

Response: We thank the reviewer for pointing out that the manuscript undermined the interests of liquid cells in biology. To response to this referee’s comment, we revised the sentences in the end of the third paragraph and the beginning of the fourth paragraph in the introduction section to emphasize the potential benefits of liquid cells for biology as following,

“...However, these sophisticated microchips require specialized holders for mounting¹⁵. Moreover, the limited accessibility of the nano-fabrication device to produce the microchips has hindered the widespread application in biological sample studies, particularly in preliminary investigations that often necessitate extensive fine-tuning of numerous parameters in the sample preparation process.”

“Nevertheless, it is nontrivial to work at ~200 K below room temperature. Samples can easily act as cold traps and attract contaminations. Achieving stability with the big temperature difference inevitably requires highly sophisticated and high-maintenance cryo-TEMs. Sample vitrification also requires elaborate machines and accessories. Moreover, it is important to note that the process of vitrification, particularly with the generation of the air-water interface, can potentially distort protein structure or lead to the dissociation of multiunit protein complexes. The ability to resolve and track protein conformations in the liquid state, close to the physiological temperature, would be a major development in structural biology highly complementary to cryo-EM.

In order to image volatile or solution samples at room temperature with TEM, ...”

The sentence

“Importantly, true projections of the biological samples in their native environments with volume information can be obtained with significantly higher contrast than in cryo-EM.”

is also added to the first paragraph of the discussion.

Comment #1.2: *Second, I find the discussion of the nanoparticles seen in Fig. 4 to be*

uninformative, particularly those that make up the dominant peak at 55 nm in the size distribution. What are they or, at least, what could they be and why do they far outnumber the viral particles? If these are not seen in cryoTEM imaging or negative stain imaging, then they must be an artifact of sample preparation. The fact that the nanoparticles seen in the images of Fig. 5., particularly the ones attached to the cell in Fig. 5a and those being related to viral protrusions in Fig. 5b., have the same diameter as the nanoparticles that make up the peak in the size distribution makes this an important point to adequately address. The paper would be strengthened if this unfortunate aspect of the samples were resolved or at least explained.

Response: We appreciate the feedback on the shortcoming in the discussion of the small particles observed with the infected HeLa cells. Extracellular vesicle release has been reported in stressed cells but their biological functions are still not fully understood (e.g. Hoen et al., PNAS 113, 9155 DOI:10.1128/JVI.00844-15 and recent articles citing this article). These vesicles may contain viral proteins, miRNA, etc. and may mediate viral spread or anti-viral response in the host.

The sentence “The more abundant smaller particles may be vesicles released by the cells during infection” is added to the legend of Fig. 4d and the discussion at the end of the paragraph describing Fig. 4 is expanded:

“The larger population of smaller particles may have originated from the extra membrane generated during virus fusion (see discussion section) or they may be extracellular vesicles released by the stressed cells important in cell-cell communication [Hoen et al.] ... Smaller particles are observed to be in the surrounding of the larger particles at these sites.”

Also, a supplementary figure (Extended Data Fig. 2) is added to show that the nanoparticles were seen only with challenged cells and not in the sample with the HeLa cell alone.

Comment #1.3: *Third, I am fairly certain the data in Fig. 5 are over interpreted. Identification of viral spikes, and membrane thickening, protein-lipid nanoparticles and uncoating of the endocytosed particles looks to me to be beyond what is a fair interpretation. The Discussion goes even further and admittedly presents a speculative model, concluding, “The proposed entry pathway differs from current conceptions of virus entry...” I am a proponent of speculating on what experimental results might apply, as long as the authors make it clear that the statements are, indeed, speculative. However, for the non-expert, the speculation presented here seems to be an extreme leap of faith. But, I am not a biologist; it may well be that an expert in cell biology and viral infectivity would look at those images and see the interpretation as obvious. I will leave it to other reviewers to make this assessment.*

Response: The reviewer is right that the interpretations are highly speculative. We have reworded the legend of Fig. 5 as well as the discussion to avoid any misunderstanding. We have also deleted the parts that involve presumptions that obviously need more supporting experiments as following,

Fig. 5: Liquid-phase TEM images (Raw), projection (3D Proj.) and 3D reconstruction of lentiviral vectors interacting with HeLa cell membrane. accompanied by a hypothetical cell entry process diagram. e, Schematic drawing illustrating a highly speculative process of lentivirus vector cell entry inspiration by the observations shown in (a-d).

In Discussion:

“The findings from our investigation of HeLa cells with lentiviral vectors have revealed potential stages of viral cell entry as illustrated in **Fig. 5a-d**. These observations have sparked the formulation of a highly speculative infection pathway for VSV-G initiated virus entry, as depicted in **Fig. 5e**. We hypothesize that the protrusions (black arrows indicated in **Fig. 5a**) could represent the viral surface spike proteins, specifically the glycoprotein VSV-G. It is known that glycoprotein VSV-G interacts with cell receptors during the process of cell entry, as exemplified by its binding to low-density lipoprotein receptors (LDL-R)⁴³. The additional structures observed (cyan arrows and orange arrow in **Fig. 5a**) could potentially correspond to cell receptors and co-receptors⁴⁴ that initiate the membrane fusion process. However, the confirmation of these speculations would obviously necessitate further experiments.

Additionally, we propose a hypothesis that the nanoparticles seen in **Fig. 5b** could be liposomes, formed by the virus membrane assimilating spike proteins during the destabilization of the cell plasma membrane as the virus-like particle penetrates. Despite ~300 nanoparticles being observed in the solution around the cell, almost none of the ~15 virus-like particles in the cellular solution were found with attached nanoparticles. In contrast, two out of three virus-like particles affixed to the cell surface were observed with attached nanoparticles, suggesting that these nanoparticles could play a role in the virus entry process. It is known that lipid rafts, which play a role in virus entry by localizing cell receptor for viral entry, can reduce the fluidity of the cell membrane⁴⁵. If the viral membrane were to merge with the plasma membrane, we would expect to observe features such as wrinkles or alterations in the membrane surface area. However, such features were not observed in our experiments. In fact, a simple estimation indicated that the fused viral/cell membrane had a similar surface area to the total surface area of the three nanoparticles (**Extended Data Fig. 3**). Therefore, we propose that liposomes are formed from the excess lipids of the viral membrane during virus entry. This hypothesis also provides an explanation for the presence of small particles in the solution, which were not observed in the solution of the HeLa cell sample without the lentivirus (**Extended Data Fig. 2**). Extracellular vesicles released by the stressed cells may also have contributed to the abundance of the small particles.

If the nanoparticles observed at the site of virus entry are indeed liposomes formed by the excess lipids resulting from the fusion of the plasma membrane and the viral membrane (**Extended Data Fig. 3**), it suggests that the matrix protein would come in direct contact with the cytoplasm upon entry, resembling virus egression. Further investigations using higher-resolution imaging techniques, such as cryo-EM with direct electron detectors, or immunolabelling, will be necessary to confirm these findings. Additionally, the potential presence of liposomes containing spike proteins, and its implications in the host immune response, warrants further investigations.”

Comment #1.4: *Finally, although the manuscript is generally well written and easy to comprehend, there are numerous minor grammatical issues that need to be addressed.*

Response: To respond to this comment, we have carefully revised the English of our manuscript as best we can.

Reviewer #2 (Remarks to the Author):

This is a manuscript by the Ren lab, focused on applying the IPET technique to biological specimens enclosed in a liquid environment. The sample preparation technique is easy to use as

described, helping to make liquid-EM studies more accessible to the community. The samples were nicely selected using single particle techniques to evaluate gro-el and lentiviruses to infect hela cells. I recommend its publication pending a few minor adjustments.

Comment #2.1: *Can the authors better define the term “molten sodium” a bit more clearly? I understand they have a publication on this technique already, but for the uninformed reader who may be looking for a concentration measurement, it would be helpful.*

Response: To response to this comment, we added following information to the related sentence as following method.

The term “molten sodium” has been changed to “liquified sodium (obtained from the electron-irradiated sodium chloride)”.

Comment #2.2: *Can a few very low magnification images be provided in figure 1 or be added to the supplemental materials for the reader to get the sense of liquid coverage and differences in liquid thickness?*

Response: Yes, low-magnification images have been added to the supplemental materials (new **Extended Data Fig. 1**).

Comment #2.3: *The zero-loss EF-TEM images look great, well done! What is the signal-to-noise ratio of representative images of gro-el in liquid compared with NS? This would help to quantify the amplitude contrast and numerically support the claims of similar contrast.*

Response: A comparison of the signal-to-noise ratios of the LCTEM sample and the NS sample is included in the **supplemental table 1**.

Comment #2.4: *In figure 2f and g, its a bit hard to see the seven domains, can they be numbered on the panels to help the reader distinguish the features?*

Response: Yes, two new panels (2h and i) are added to Figure 2 to help the readers to distinguish the domains in the ‘top’ and ‘side’ views.

Comment #2.5: *Have you thought about doing 3D reconstructions of the gro-el data? Seems like you would achieve a nice structure....please comment whether you think this is possible, or why it may not be feasible (limited views, etc)...*

Response: Yes, we have. However, 3D reconstruction would not be satisfactory with the present data. The resolution achieved in our experiments was limited by our instruments (Zeiss 120 Libra TEM with LaB6 filament, Gatan Ultrascan CCD). We also think that image formation in liquid cell TEM (e.g. multiple scattering in the liquid phase) needs to be better understood for meaningful interpretation of high-resolution structures.

Comment #2.6: *Is the liquid thickness in these enclosures dictated by the diameter of the sample particles? If so, what’s the estimated thickness of each type of sample (particles vs viruses)...I like the virus entry studies. Albeit speculative, it brings a new technique to the field that one can optimize for future lines of investigation.*

Response: We thank the reviewer for the positive comments. Indeed, the liquid thickness is a function of the sample. As seen in the new supplementary figures (Extended Data Fig. 1-2), the thickness of the GroEL grid is quite homogeneous and thin whereas the cell sample has quite a large thickness range. The regions with cells are much thicker than the regions with only the

vesicles in the cell sample, showing that, to some extent, the formvar films adapt to the local sample thickness. Wrinkle or curvature of the formvar film can also cause local variation in sample thickness.

Comment #2.7: *Minor technical note, I was unable to open the video files, but I did see the supplemental materials.*

Response: The videos are included as separate files in the supplementary materials, each in the .avi format, in accordance with the journal's requirements.

Reviewer #3 (Remarks to the Author):

This reviewer would mainly like to comment on the virus entry aspect of this manuscript. It is an impressive feat that authors could observe the change in morphology of the virus during infection using the liquid cell system. However, the authors' proposed detailed description of the entry process is highly speculative and contradictory to the current consensus in the field on the viral entry process. The authors should tone down the biological claims significantly until they have very strong evidence – immune-staining, statistics etc. Without additional experiments, the authors should just describe the general viral entry process, and not point to any proteins and receptors or speculate in detail about mechanisms of viral entry etc.

Major comments:

Comment #3.1: *The authors comment on the spike and M proteins by pointing out negative (white) densities, but if those are supposed to be proteins, so why would they have negative densities? It is probable that the negative densities are bubbles caused by radiation damage to the sample, and thus it is best not to make claims like this. The contrast of the sample relative to the background depends on the relative density and the scattering capacity. As in the case of the GroEL, the small proteins have negative densities because the solid protein scatter less than the liquid.*

Response: The reviewer is correct that the liquid cell samples are very sensitive to radiation damage and we should definitely be wary of any signs of artifacts introduced by the electron beam. However, we are confident that these features were not bubbles caused by radiation damage because they were stable during the tilt series acquisition and had the same appearance in the first images (low-dose) as in the last images (high-dose). The reconstructed tomogram also showed consistent particle size and shape without smearing, which would have been the case if these were expanding bubbles from radiation damage.

As observed by the reviewer, the ‘negative’ densities in the GroEL sample, or their higher contrast, come from the fact that “*the solid protein scatters less than the liquid*”. Similarly, these little protrusions must either be less dense or scatter less than their neighboring features but do not necessarily imply that they are bubbles.

Comment #3.2: *A lot of these observational claims must be supported by statistics, e.g. how often do the liposomes associated with viruses appear during entry or they are just random?*

Response: we agree with the reviewer that it is difficult to be certain about the origin of these small particles without further complementary experiments. We revised the sentence by added statistics in the discussion section as below,

“Additionally, we propose a hypothesis that the nanoparticles seen in **Fig. 5b** could be liposomes, formed by the virus membrane assimilating spike proteins during the destabilization of the cell plasma membrane as the virus-like particle penetrates. Despite ~300 nanoparticles being

observed in the solution around the cell, almost none of the ~15 virus-like particles in the cellular solution were found with attached nanoparticles. In contrast, two out of three virus-like particles affixed to the cell surface were observed with attached nanoparticles, suggesting that these nanoparticles could play a role in the virus entry process...”

Comment #3.3: *The authors speculate that the liposomes are parts of the viral membrane being released during the virus entry process by comparison to uninfected cells. However, infected cells are generally sicker than uninfected cells, with a more variable morphology as a result. How would the authors know if the liposomes are derived from the virus and not shedding of membrane from the sick cells?*

Response: We thank the reviewer for pointing out the alternative origins of the small particles. We have included this possible interpretation in the text in the discussion of Figure 4:

“The larger population of smaller particles may have originated from the extra membrane generated during virus fusion (see discussion section) or they may be extracellular vesicles released by the stressed cells important in cell-cell communication [Hoen et al., PNAS 113, 9155 DOI:10.1128/JVI.00844-15].”

Comment #3.4: *The authors should provide statistics about the claimed thick cell membranes and their localization in regions around virus and not around virus.*

Response: We have included statistics of the intensity across the cell membrane in regions around virus and not around virus in the new Extended Data Fig. 4.

Comment #3.5: *Statistics are required for how often conical particles are associated with globular particles (capsid and matrix, respectively, as claimed by the authors) post-viral entry.*

Response: We have included particles located inside the cells, which we judged to be viral particles, in the new Extended Data Fig. 8. There were about the same number of particles with and without the attachment. We agree absolutely with the reviewer that more statistics is required to confirm this observation.

Comment #3.6: *This reviewer thinks that the authors should focus on the liquid cell method and showing that it is applicable to the study of both purified protein samples and cells infected with viruses. A deeper examination of speculated entry mechanisms based on the data in the manuscript is out of the scope of this manuscript and could be highly controversial and would need to be supported by orthogonal experiments that were not shown here.*

Response: We have heeded the reviewer’s advice and tone down the biological claims by deleted most of the speculations in the legend of Fig. 5 and in the discussion and emphasized the necessity of further complementary experiments.

REVIEWERS' COMMENTS

Reviewer #1 (Remarks to the Author):

The manuscript by Kong et al. has been revised to adequately address my concerns. I have one final optional suggestion for the authors. In my first comment, I recommended that the authors better articulate the value of the work beyond creating a "poor man's" version of cryoTEM. The authors have improved introduction to achieve this. However, there is no mention of the fact that their results lead them to a somewhat novel hypothesis about the process of cell entry. My recommendation is that they point this out at the end of the introduction so that the reader knows the results are more than a demonstration of the imaging power of the method. Specifically, after the sentence, "By capturing snapshots of viral-cell entry in this system, we established that our technique could be applied to the studies of short-lived biological processes" I recommend adding something along the lines of. "Moreover, the results obtained here lead to novel hypotheses concerning the process of cell entry."

Reviewer #3 (Remarks to the Author):

Comment #3.1

Response: The reviewer is correct that the liquid cell samples are very sensitive to radiation damage and we should definitely be weary of any signs of artifacts introduced by the electron beam. However, we are confident that these features were not bubbles caused by radiation damage because they were stable during the tilt series acquisition and had the same appearance in the first images (low-dose) as in the last images (high-dose). The reconstructed tomogram also showed consistent particle size and shape without smearing, which would have been the case if these were expanding bubbles from radiation damage.

As observed by the reviewer, the 'negative' densities in the GroEL sample, or their higher contrast, come from the fact that "the solid protein scatters less than the liquid". Similarly, these little protrusions must either be less dense or scatter less than their neighboring features but do not necessarily imply that they are bubbles.

New comments by reviewer:

Reviewer accept the explanation on the radiation damage.

However, below should be address or claims be toned down:

In Figure 5a (left), the authors put arrows on densities that are negative densities (bright white spots)- but from the core (center part) of the virus, you can tell the higher the protein densities the darker the pixels and so protein densities should be in a gradient of different "darkness" of the pixel. The bright negative densities could due to the CTF effects, and thus the authors are pointing to the wrong things - they are not proteins.

Comment #3.2: A lot of these observational claims must be supported by statistics, e.g. how often do the liposomes associated with viruses appear during entry or they are just random?

Response: we agree with the reviewer that it is difficult to be certain about the origin of these small particles without further complementary experiments.

We revised the sentence by added statistics in the discussion section as below,

"Additionally, we propose a hypothesis that the nanoparticles seen in Fig. 5b could be liposomes, formed by the virus membrane assimilating spike proteins during the destabilization of the cell plasma membrane as the virus-like particle penetrates. Despite ~300 nanoparticles being observed in the solution around the cell, almost none of the ~15 virus-like particles in the cellular solution were found with attached nanoparticles. In contrast, two out of three virus-like particles affixed to the cell surface were observed with attached nanoparticles, suggesting that these nanoparticles could play a role in the virus entry process..."

New comments by reviewer:

Your statement says you have 3 virus-like particles that are attached to a cell and 2 of them show the nanoparticles present. This is not statistically significant. The nanoparticles could be associated with the cells, not virus.

REVIEWERS' COMMENTS

Reviewer #1 (Remarks to the Author):

Common #1.1: *The manuscript by Kong et al. has been revised to adequately address my concerns. I have one final optional suggestion for the authors. In my first comment, I recommended that the authors better articulate the value of the work beyond creating a “poor man’s” version of cryoTEM. The authors have improved introduction to achieve this. However, there is no mention of the fact that their results lead them to a somewhat novel hypothesis about the process of cell entry. My recommendation is that they point this out at the end of the introduction so that the reader knows the results are more than a demonstration of the imaging power of the method. Specifically, after the sentence, “By capturing snapshots of viral-cell entry in this system, we established that our technique could be applied to the studies of short-lived biological processes” I recommend adding something along the lines of. “Moreover, the results obtained here lead to novel hypotheses concerning the process of cell entry.”*

Response #1.1: To response to the reviewer’s recommendation, we have extended the last sentence in the introduction as following,

The last sentence in the introduction section:

“By capturing snapshots of viral cell entry in this system, we establish that our technique can be applied to the studies of transient biological processes, which may lead to insights and hypotheses concerning important biological activities.”

Reviewer #3 (Remarks to the Author):

Comment #3.1: Reviewer accept the explanation on the radiation damage. However, below should be address or claims be toned down:

In Figure 5a (left), the authors put arrows on densities that are negative densities (bright white spots)– but from the core (center part) of the virus, you can tell the higher the protein densities the darker the pixels and so protein densities should be in a gradient of different “darkness” of the pixel. The bright negative densities could due to the CTF effects, and thus the authors are pointing to the wrong things – they are not proteins.

Response #3.1: The images shown have been corrected for their CTF. We understand that the term ‘negative density’ may be confusing to readers and have included an additional phrase to explain the term:

The three sentences in the section entitled “IPET of virus cell entry”:

“..., Fig. 5 presents magnified views of the particles obtained from the zero-tilt projection of this sample region, as well as the projections derived from the 3D reconstruction and its density maps (the superposition of the positive and negative density maps, which highlight features that scatter electrons more and less than the background, respectively), following data processing with IPET (Supplementary Fig. 5-7). The resolution of the maps, estimated by the Fourier Shell Correlation (FSC) of 0.5, showed that the maps had a resolution better than 10 nm (Supplementary Fig. 5-7).”

Comment #3.2: Your statement says you have 3 virus-like particles that are attached to a cell and 2 of them show the nanoparticles present. This is not statistically significant. The nanoparticles could be associated with the cells, not virus.

Response #3.2: We have followed the reviewer and the editor's advice and clearly stated the limit of our studies in the revised manuscript:

The fourth paragraphs in the discussion section:

“The preliminary findings from our investigation of HeLa cells with lentiviral vectors have revealed potential stages of viral cell entry as illustrated in Fig.5a-d. While emphasizing the fact that the present dataset is small and preliminary dataset, which is statistically insignificant for any appropriate claim, we formulated a highly speculative infection pathway (depicted in Fig. 5e) for VSV-G initiated virus entry based on these preliminary observations as a proof-of-principle of what the technique may be capable of delivering in cellular EM. We hypothesize that the protrusions (black arrows indicated in Fig. 5a) could conceivably represent the viral surface spike proteins.”

with the phrase “, specifically the glycoprotein VSV-G” deleted.

The last sentences in the last third paragraphs in the discussion section:

“... Therefore, we propose that liposomes are formed from the excess lipids of the viral membrane during virus entry. Whereas extracellular vesicles released by the stressed cells may have contributed to the main population of the small particles in the solution, which was not observed in the solution of the HeLa cell sample without the lentivirus (Supplementary Fig. 2), this hypothesis suggests that nanoparticles from excess viral membrane may also have contributed to the abundance of small particles. Further experiments to collect a statistically significant data set will be essential to verify the hypothesis.”

The last paragraph in the discussion section:

“The study showcases the potential of a straightforward hermetic TEM grid preparation method as a valuable and cost-efficient technique for molecular imaging of hydrated biological samples at ambient temperature. This approach complements other microscopy techniques and presents a promising solution for exploring crucial biological questions at the molecular level. Although our current observations are insufficient to draw any definitive biological conclusions, the demonstration of the potential of this methodology serve as a proof-of-principle on its prospective applications.”